# Quantitative Three-Dimensional Color Power Angiography Parameters Predict Response to Locally Injected Bleomycin in Infantile Hemangioma

**DOI:** 10.3390/diagnostics15222903

**Published:** 2025-11-16

**Authors:** Kai Gu, Yi Zhong, Jiexin Wen, Zhaoxia Wang

**Affiliations:** 1Department of Orthopedics, Chongqing Key Laboratory of Pediatrics, Ministry of Education Key Laboratory of Child Development and Disorders, National Clinical Research Center for Child Health and Disorders, China International Science and Technology Cooperation Base of Child Development and Critical Disorders, Children’s Hospital of Chongqing Medical University, Chongqing 400014, China; igk_work@126.com; 2Department of Ultrasonography, Chongqing Key Laboratory of Pediatrics, Ministry of Education Key Laboratory of Child Development and Disorders, National Clinical Research Center for Child Health and Disorders, China International Science and Technology Cooperation Base of Child Development and Critical Disorders, Children’s Hospital of Chongqing Medical University, Chongqing 400014, China; yi_zhong2025@hospital.cqmu.edu.cn

**Keywords:** ultrasonography, infantile, hemangioma, bleomycin, injection

## Abstract

**Background**: This study aimed to explore the key factors affecting the efficacy of local bleomycin injection by quantitatively analyzing the blood flow parameters of lesions before treatment using three-dimensional color power angiography to allow early prediction of the treatment response in patients with infantile hemangioma. **Methods**: The clinical data of children diagnosed with infantile hemangioma and treated with locally injected bleomycin at the Children’s Hospital of Chongqing Medical University between February 2018 and October 2024 were retrospectively analyzed. The treatment efficacy grades of the included patients were determined. Baseline clinical information and the three-dimensional blood flow parameters (the vascularization index, flow index, and vascularization–flow index) were collected and compared with the treatment response. The predictive ability of the identified indicators was evaluated using receiver operating characteristic curves. **Results**: Of the 40 patients included in the study, 13, 16, 8, and 3 had treatment efficacy grade I, II, III, and IV lesions, respectively. There were no significant differences in age or lesion location, type, or volume before treatment among lesions of different treatment efficacy grades (*p* > 0.05). The vascularization index, flow index, and vascularization–flow index were significantly higher in grade I lesions than in those of the other three groups (*p* < 0.001, *p* = 0.008, and *p* = 0.003, respectively), and these indices decreased as the efficacy grade increased. The vascularization–flow index was identified as an independent predictor of treatment efficacy, with a sensitivity of 81.5%, a specificity of 84.6%, and an area under the receiver operating characteristic curve of 0.85 at the optimal cutoff value of 23.3. **Conclusions**: The vascularization–flow index can be used as an objective predictor of the response of infantile hemangioma to locally injected bleomycin. It is recommended that patients with a vascularization–flow index <23.3 receive locally injected bleomycin, and those with a vascularization–flow index ≥23.3 should receive an alternative treatment option.

## 1. Introduction

The incidence of infantile hemangioma (IH) is approximately 5% among live births, making it the most common benign soft tissue tumor in pediatric dermatology. Its pathogenesis is associated with the abnormal proliferation of vascular endothelial cells and neovascularization [1,2]. Although most IH cases are self-limiting, some patients may develop complications such as ulcers, bleeding, functional disorders or disfigurement. In up to 20% of cases, early intervention may be strongly considered. This not only brings significant psychological stress and economic burden to the families of the affected children, but also consumes a large amount of medical resources [3,4]. Moreover, there may be racial disparities in the incidence of IH; for instance, it is more commonly observed in Caucasian infants, highlighting an inequitable distribution of the disease burden [5]. Thus, the effective management of IH underscores a matter of significant public health implications.

Among the various treatment methods for IH, the effectiveness and practicality of local percutaneous injections of bleomycin have been clinically recognized [4,6,7]. Bleomycin injected into the lesion is therapeutically effective and greatly reduces the incidence of adverse reactions caused by systemic treatment. In addition, injections of bleomycin are required only once a month, contributing to its widespread acceptance among clinicians and making it the preferred treatment of pediatric patient families [6,7,8,9].

We have previously studied the maximum safe and effective dose of bleomycin for the treatment of IH, with the aim of optimizing its therapeutic effects [7]. However, treatment efficacy varies among lesions, hindering early treatment and increasing the risk of complications. Insufficient treatment of high-risk lesions may even be life-threatening, and in medium- and low-risk lesions, the skin appearance may not return to normal owing to incomplete tumor regression, which may cause mental health issues [4]. Screening of lesions before initiating treatment would allow alternative treatment options to be selected for lesions not sensitive to local injection therapy.

However, the exact reason for the insensitivity of certain IH lesions to local injection therapy has not yet been clarified [7,10]. Previous studies have positively correlated two-dimensional Doppler ultrasound blood flow parameters such as vascularity increase, vascular resistance index, and pulsatility index with the efficacy of oral propranolol in the treatment of IH [11,12]. This indicates that the characteristics of blood vessels and blood flow are important factors influencing therapeutic efficacy. Therefore, we hypothesize that the proportion of vascular space and the intensity of the blood flow vector in IH lesions may be key factors influencing bleomycin treatment response variation.

Three-dimensional color power angiography (3D-CPA) comprehensively reflects blood flow within a lesion. Reconstructed three-dimensional (3D) images reveal the entire hemangioma, and can be used to calculate three important parameters: the vascularization index (VI), flow index (FI), and vascularization–flow index (VFI). Thus, blood perfusion in hemangiomas can be quantitatively evaluated from different dimensions of vascular proportion and blood flow intensity [7,13]. To test our hypothesis, 3D-CPA was used to image lesions in patients with IH, to identify the key factors affecting the efficacy of local injection therapy. The aim of the study was to allow early prediction of the response of IH lesions to local injection therapy, to provide a more accurate basis for clinical decision making.

## 2. Materials and Methods

### 2.1. Patients

This study retrospectively analyzed the data of patients with IH diagnosed [14] and treated with local injections of bleomycin at the Children’s Hospital of Chongqing Medical University between February 2018 and October 2024. The inclusion criteria were as follows: (1) maximum diameter of a single lesion < 5 cm, (2) 3D-CPA performed before treatment was initiated and generated clear images, (3) 3D-CPA performed one month after treatment was initiated and generated clear images, and (4) no previous treatment. The exclusion criteria were as follows: (1) treatment withdrawn due to serious adverse reactions or subjective reasons, and (2) non-compliance with treatment or follow up. Finally, 40 children were included in this study.

This study was approved by the Ethics Committee of the Children’s Hospital of Chongqing Medical University (Ethics Review Research no. 058 in 2025). Written informed consent was obtained from the parents of all the patients.

### 2.2. Patient Management

Detailed medical histories were taken and physical examinations were conducted after the children presented at the hospital. Patient age and the type, location, and growth of the lesions were recorded. Before treatment initiation, 3D-CPA was performed and the parents of the patients were informed of the treatment plan, which was conducted in an outpatient setting. Once a month, 0.3–0.4 mL/cm^3^ of a 0.15% bleomycin solution (Hanhui Pharmaceutical Co., Ltd., Hangzhou, Zhejiang, China) was injected into lesions, with the maximum dose not exceeding 3 mg [7,15]. The needle was inserted from the edge of the hemangioma and the drug solution was injected radially. It was confirmed that the withdrawal syringe was free of blood before injection. The injection was typically stopped when the skin of the hemangioma became mildly whitened. The children were reevaluated using 3D-CPA one month after the first injection (Figure 1).

### 2.3. D-CPA

Imaging was performed while children were quiet or sleeping owing to pacification, feeding, natural sleep, or sedation. The examination was performed using a EPIQ5 Ultrasound system (Philips, Amsterdam, The Netherlands) with a VL13-5 3D volumetric probe and fixed settings suitable for small organs (range: 700 MHz; frame rate: 42%; angle: 30°). The 3D mode was initiated at the maximum section of the lesion, and the images were saved after obtaining satisfactory scanning information using the 3D gray scale and 3D-CPA mode. The procedure was repeated three times under the same conditions.

Image analysis was performed using QLAB Advanced Ultrasound Quantification (Philips). The 3D volume and blood flow parameters of each lesion were calculated. The 3D blood flow parameters VI, FI, and VFI were calculated as follows: VI = color voxels/(total voxels − background voxels), FI 1 = weighted color voxels/(color voxels − border voxels), FI 2 = weighted color voxels/color voxels, VFI 1 = weighted color voxels/(total voxels − background voxels), and VFI 2 = weighted voxels × cube volume/(cyst volume × total voxels) (Figure 2) [13].

### 2.4. Assessment of Treatment Effect

The Achauer efficacy evaluation method [7,16] was used to assess the therapeutic effects of bleomycin injections (Table 1). The volume of the lesion measured by three-dimensional ultrasonography was used to evaluate the change in the tumor volume. The color changes on the tumor surface were evaluated by comparing with the digital photos collected before treatment.

The three-dimensional ultrasound image analysis was carried out by two ultrasound physicians with a minimum of three years of professional experience. These physicians were tasked with the post-processing of the images. Subsequently, the software computed and generated the three-dimensional parameters of the lesion. These parameters included the volume value V (in milliliters), as well as VI, FI, and VFI. The clinical medical team was composed of two clinical doctors with over three years of experience in the diagnosis and treatment of infantile hemangiomas. They assess the alterations in the color of the tumor surface. This assessment was achieved through a comparison with the digital photos taken prior to treatment for comprehensive evaluation.

All the analysts received unified training on evaluation standards before conducting the analysis. They independently completed all the image analyses and efficacy assessments. Throughout the process, strict blinding was implemented. All ultrasound images and digital photos of the tumors were anonymized and randomized. Neither the two groups of analysts nor the other evaluators were aware of the patients’ treatment information or the conclusions of other evaluators.

### 2.5. Statistical Analysis

SPSS software version 26.0 (IBM Corp., Armonk, NY, USA) was used for data analysis. Measurement data are presented as mean ± standard deviation. Statistical significance was set at *p* < 0.05. Normality analysis was performed using the Shapiro–Wilk normality test. Levene’s test was used to analyze the homogeneity of variance. The chi-squared test was used to compare differences in age at initial diagnosis, lesion location, and lesion type among patients with different efficacy grades. Univariate logistic regression was used to analyze the differences in the baseline 3D-CPA parameters among patients with different efficacy grades. Redundant variables were removed using a stepwise forward regression method, and predictors were identified. The value that maximized the sum of sensitivity and specificity was selected as the optimal cutoff value for the predictor. Receiver operating characteristic curves were generated to evaluate the predictive ability of the indicators.

## 3. Results

### 3.1. Clinical Features of All Patients of IH

A total of 40 patients received local injection treatment. The cohort consisted of 29 women (72.5%) and 11 men (27.5%). The age distribution at the initial diagnosis was as follows: 16 patients (40.0%) were 0–3 months old; 13 cases (32.5%), 4–6 months old; 4 cases (10.0%), 7–9 months old; and, 7 cases (17.5%), 10–12 months old. The trunk was the most common location of the lesion, with 22 cases (55.0%), followed by the face with 9 cases (22.5%), the head and neck with 6 cases (15.0%), and the limbs with 3 cases (7.5%). In terms of lesion type, the majority were mixed types (31 cases, 77.5%) and deep types (9 cases, 22.5%). No superficial type lesions were observed in this cohort. Among the 40 children, 8 cases (20.0%) had mild local swelling, 5 cases (12.5%) experienced brief low fever after treatment, and all children had no serious complications.

### 3.2. Assessment of Treatment Efficacy

According to the Achauer efficacy evaluation method, treatment efficacy one month after the first injection was grade I in 13 patients, grade II in 16 patients, grade III in 8 patients, and grade IV in 3 patients. Representative 3D-CPA images of patients with grade I and IV treatment efficacy are shown in Figure 3 and Figure 4, respectively.

### 3.3. Baseline Lesion Characteristics According to Treatment Efficacy

No significant differences were observed in the age, lesion location, or lesion type prior to treatment of patients with different treatment efficacies (*p* > 0.05) (Table 2).

### 3.4. Baseline 3D-CPA Parameters of Lesions According to Treatment Efficacy

No significant differences were observed in baseline lesion volume among the different treatment efficacy grades (*p* = 0.061) (Table 3). However, the 3D-CPA blood flow parameters VI, FI, and VFI were all highest in patients with grade I treatment efficacy and decreased as the treatment efficacy grade increased; the differences among the groups were statistically significant (*p* < 0.001, *p* = 0.008, and *p* = 0.003, respectively) (Table 3 and Figure 5).

### 3.5. Predictors of Treatment Response

Univariate logistic regression analysis was used to evaluate the associations between the 3D-CPA blood flow parameters VI, FI, and VFI and treatment efficacy. The results showed that the VI, FI, and VFI were all significantly correlated with treatment efficacy in patients with IH receiving local bleomycin injections (Table 4). We used the variance inflation factor (VIF) to test for multicollinearity. The VIF of VI was 4.96; FI, 1.77; and, VFI, 5.10 (all VIF values were less than 10).

Following multivariate logistic regression analysis of variables determined using forward selection, only the VFI (odds ratio = 0.818, 95% confidence interval: 0.705–0.948, *p* = 0.008) remained an independent predictor of treatment efficacy.

### 3.6. Optimal VFI Cutoff Value

The optimal cut-off value of VFI for identifying children with poor treatment response in the first-level efficacy group was determined to be 23.3 using the receiver operating characteristic curve analysis, with the maximization of the Youden index as the criterion. Receiver operating characteristic curve analysis of the ability of the VFI to predict treatment response identified an optimal cutoff value of >23.3, with a sensitivity of 81.5%, a specificity of 84.6%, and an area under the receiver operating characteristic curve of 0.85 (Figure 6).

## 4. Discussion

IH is the most common soft tissue benign tumors in infants, with a natural history characterized by rapid proliferation and slow regression [17]. Treatment in the early stages of proliferation, in which vascular endothelial cells are highly active and hemangiomas grow rapidly, is therefore key. It is essential to effectively inhibit endothelial cell proliferation and angiogenesis through various therapeutic approaches before the hemangioma reaches its maximum volume. Effective early intervention controls tumor growth [4,18,19,20] and reduces the risk of complications associated with rapid hemangioma expansion, such as ulceration, hemorrhage, and infection. Early treatment can also promote earlier regression of the hemangioma and shorten the course of IH. It reduces the risk of residual hemangioma, scarring, and compression and invasion of surrounding tissues, thereby decreasing the likelihood of dysfunction and appearance alteration [4,19]. Therefore, early intervention is particularly important for the treatment of IH [20].

Early intervention requires the timely selection of treatment modalities that effectively inhibit vascular endothelial cell proliferation and have good safety profiles [2,4,19]. Current treatment methods for IH include locally injected drugs, oral drugs, laser therapy, surgical resection, and radiological intervention [7]. Locally injected drugs have shown unique advantages in clinical practice owing to their precise action on target lesions and low systemic exposure. The use of locally injected drugs for the treatment of IH was first reported in the late 20th century [21], and medications including glucocorticoids and lauromacrogol have been used in this way [22,23]. In recent years, bleomycin has become an important choice for the treatment of IH by local injections [6,7,8,9]. Bleomycin has been widely used to treat a variety of malignant tumors because of its unique anti-angiogenic and cytotoxic effects. Although bleomycin treatment of malignant tumors has been associated with significant side effects [15], no serious systemic toxic effects have been observed following local injection into hemangiomas. Local swelling and pain have been reported in a minority of patients but typically resolve within 24–48 h [15]. Therefore, local injection of bleomycin has a good safety profile for the treatment of IH [7,15].

In the present study, all hemangiomas treated with bleomycin reduced to varying degrees one month after the first injection. These results indicated that local injection of bleomycin can inhibit the proliferation of IH. This is probably owing to the ability of bleomycin to promote the apoptosis of vascular endothelial cells, reduce neovascularization, and ultimately inhibit hemangioma growth by specifically binding to vascular endothelial cell DNA, inhibiting the expression of vascular endothelial growth factor, and regulating the p53/B-cell lymphoma-2 pathway [3,8,24,25,26,27].

This study chose one month after the first treatment as the main time point for evaluating the therapeutic effect based on the following reasons: first, there is evidence supporting the predictive value of this early time point. For instance, Yun et al.’s study clearly set the first efficacy assessment for IH at one month after the injection treatment. They found that patients with the better efficacy at one month after treatment maintained a better response before the treatment was terminated, whereas patients with the worse efficacy at one month did not show any further regression [28]. This indicates that the early response can effectively predict the long-term outcome. Early assessment can effectively identify the responsiveness to treatment. It provides a basis for timely adjustment of the treatment plan. Secondly, this measure aligns with the urgent needs of clinical practice. Delaying the assessment may lead to the continued progression of the lesion under ineffective treatment, as the proliferative hemangiomas progress rapidly. Conducting the assessment one month after the treatment is aimed at early identification of non-responders and timely intervention. This aligns with the principle emphasized in the Clinical Practice Guideline for the Management of Infantile Hemangiomas for rapidly proliferating lesions [14]. This helps to minimize the risks of complications such as ulcers to the greatest extent. Finally, this time point has good clinical feasibility and consensus. Setting the assessment at one month after treatment (or 4–6 weeks) is a common practice in the field of vascular diseases. Multiple clinical studies [4,7,29,30] have adopted a first efficacy assessment at 4–6 weeks after treatment, ensuring that the study design aligns with real-world scenarios.

As previously described, the local injection of bleomycin inhibited IH proliferation in the present study; however, significant heterogeneity in treatment response was observed. Treatment efficacy was not associated with the conventional clinical parameters assessed, suggesting that the biological characteristics of hemangiomas may be more likely to determine treatment response. Previous histopathology findings have shown that although proliferative hemangiomas typically have common characteristics, such as abnormal endothelial cell proliferation, increased microvessel density, and a relative lack of interstitial components [31,32], the spatial distribution characteristics of the vascular network and blood flow within the hemangioma vary significantly. These variations lead to significant differences in the vascular occupancy ratio and flow vector intensity of the tumors, which directly affects the pharmacokinetic behavior of bleomycin in the lesion, potentially influencing treatment efficacy.

In the present study, 3D-CPA, which quantitatively reflects blood perfusion characteristics through the VI, FI, and VFI, was used to evaluate the vascular occupancy ratio and blood flow vector intensity in IH lesions. The VI represents the blood vessel density per unit volume, the FI reflects the blood flow intensity determined by the speed of red blood cell movement, and the VFI integrates the two. These three parameters can be used to comprehensively evaluate the intensity and distribution of the blood supply to the lesion [13]. The results of the present study showed that the baseline VI, FI, VFI were significantly higher in lesions in the grade I efficacy group than in the grade II, III, and IV efficacy groups, and decreased as efficacy grade increased. This suggests that the level of blood perfusion in the lesion before treatment is closely related to the therapeutic effect of bleomycin. The higher VI, FI, and VFI in the grade I efficacy group indicated that lesions with poor treatment efficacy had a higher vascular density and blood flow intensity prior to treatment.

The poor response to bleomycin treatment of lesions with high blood flow parameters may probably involve multiple mechanisms (Figure 5). First, a high VI or VFI indicates that the lesion has a high vascular density and large blood flow, which may probably cause the dilution of locally injected bleomycin under a rapid blood flow washout, resulting in an early reduction in drug concentration. Second, the wall shear stress generated by the rapid blood flow corresponding to a high FI may not only shorten the drug–endothelial contact time but also activate the phosphoinositide 3-kinase/AKT signaling pathway to promote the growth of vascular endothelial cells [33,34] and antagonize the DNA damage caused by bleomycin. The synergistic effects of these factors may weaken the therapeutic effects of bleomycin in lesions with high blood flow. In contrast, lesions with low blood flow are more likely to respond to bleomycin treatment through the apoptosis and fibrosis of vascular endothelial cells owing to low angiogenic activity and relatively high drug concentration and residence time.

The results of the logistic regression analyses suggest that the VFI can effectively identify patients who will exhibit a poor response to locally injected bleomycin and provide an important reference for the early clinical screening of lesions to select appropriate treatment options. The predictive power of the VFI may be closely related to its characteristics as a comprehensive parameter. The VFI integrates both vessel density and blood flow velocity information to comprehensively reflect the hemodynamic status of the lesion [13]. Previous studies have shown that the therapeutic effects of propranolol on hemangiomas rely on the dual regulation of abnormal vascular proliferation and hemodynamics [11,12,35]; blood flow velocity or vessel density alone may not fully capture the complex biology of the lesion as they may be affected by individual differences or insufficient measurement sensitivity [12]. Therefore, the VFI as a multidimensional indicator is more suitable for evaluating the hemodynamic status of hemangiomas. A VFI > 23.3 indicates that the lesion has a higher vascular density, greater blood flow intensity, and more active microcirculation, which may limit the therapeutic efficacy of locally injected bleomycin. Therefore, alternative treatment modalities such as oral propranolol are recommended for these lesions. Previous studies have suggested that oral propranolol is more effective in lesions with increased vascularity according to color Doppler ultrasonography performed prior to treatment [11].

This study had certain limitations. Although hemodynamic parameters such as the VFI can reflect the macroscopic mechanical characteristics of vascular structures, they cannot fully characterize the complex biological characteristics of lesions. The present study did not systematically address components of the vascular endothelial extracellular matrix, changes in pericyte contractile function, and the status of the immune microenvironment. These biological variables may affect the distribution of bleomycin in lesions by regulating its local penetration efficiency and metabolic rate. Future studies should integrate multidimensional indicators, including extracellular matrix quantitative analysis, inflammatory factor detection, and hemodynamic parameters, to build a comprehensive prediction model that fully analyzes the differential mechanisms of IH response to locally injected bleomycin. Another limitation of this study is the potential inter-observer differences in color assessment. Although we have taken measures such as training and standardizing the shooting process to maximize consistency, this subjective factor cannot be completely avoided. This study is essentially a single-center, retrospective study with a limited sample size. Therefore, when applying the conclusions of this study to broader clinical practice or different patient groups, caution should be exercised. In the future, multi-center, prospective studies and external validation in larger samples are needed to confirm its universality.

In conclusion, this study verified the hypothesis that the vascular occupancy ratio and flow vector strength are key factors in IH treatment responses using 3D hemodynamic analysis. The VFI was identified as a potential objective predictor of the bleomycin treatment response in IH. It is recommended that patients with a VFI < 23.3 receive locally injected bleomycin, and those with a VFI ≥ 23.3 should receive an alternative treatment option. This study identified the VFI as a novel predictor that can be measured early and noninvasively to indicate the response of IH to locally injected bleomycin. The VFI has the additional advantages of being simple to measure and objective. This study provides a reference for the development of individualized treatment strategies for IH.

## Figures and Tables

**Figure 1 diagnostics-15-02903-f001:**
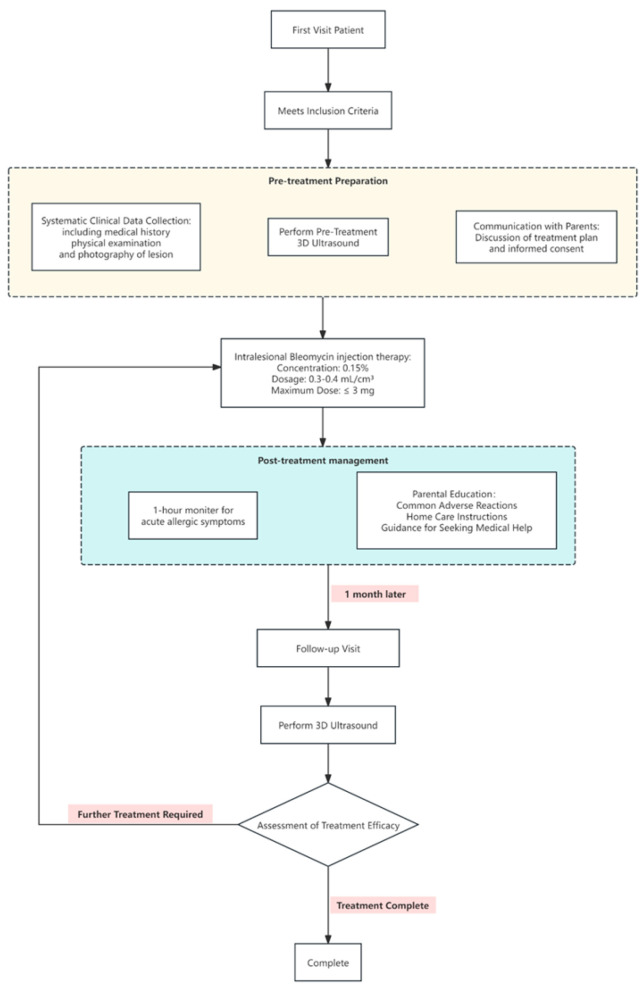
Patient treatment strategy and safety monitoring information flowchart.

**Figure 2 diagnostics-15-02903-f002:**
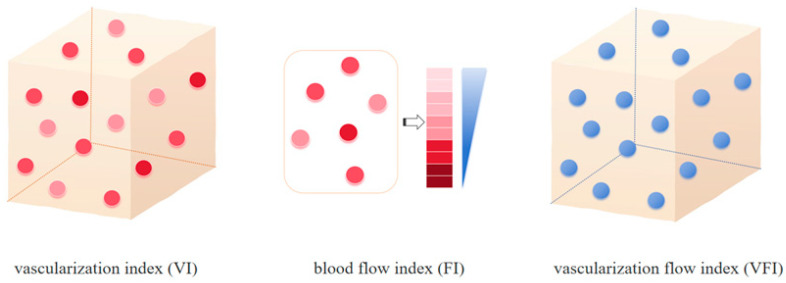
Patterns of the 3D-CPA parameters. VI refers to the ratio of color voxels to the total number of valid voxels, indicating the degree of vascularization in the tissue. FI is calculated as the ratio of weighted color voxels to the total number of color voxels, representing the average intensity of blood flow in the region during scanning and serving to quantify local blood flow intensity. VFI is defined as the ratio of weighted color voxels to the total number of valid voxels. It represented the sum of blood flow and vascularization information to comprehensively assess tissue perfusion levels [13].

**Figure 3 diagnostics-15-02903-f003:**
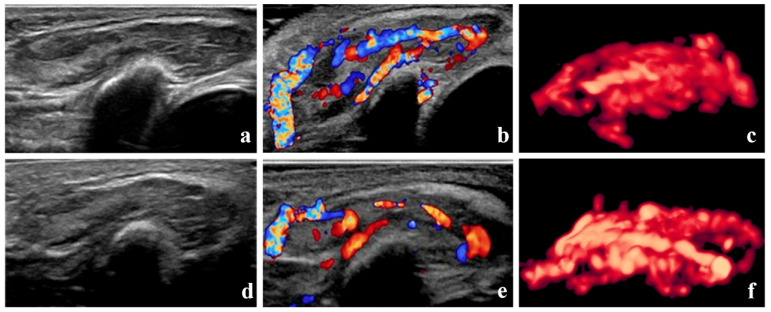
Grade I therapeutic efficacy—B-mode images, color Doppler images, and three-dimensional angiographic visualization of infantile hemangioma before and after treatment. The volume of the hemangioma was 4.2 mL before treatment and 3.9 mL one month after the initial local injection. (**a**–**c**) Before treatment, dense blood vessels can be seen in the hemangioma. The vascularization index, flow index, and vascularization–flow index were 53.5%, 44.6, and 23.9, respectively. (**d**–**f**) One month after treatment, blood vessels in the hemangioma remain dense.

**Figure 4 diagnostics-15-02903-f004:**
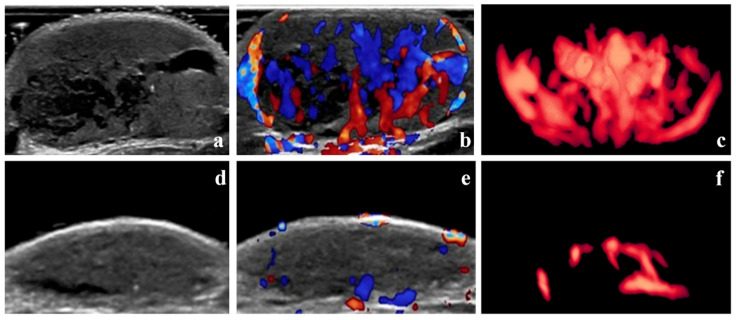
Grade IV therapeutic effect—B-mode images, color Doppler images, and three-dimensional angiographic visualization of infantile hemangioma before and after treatment. The volume of the hemangioma was 8.53 mL before treatment and 1.33 mL one month after the initial local injection. (**a**–**c**) Before treatment, dense blood vessels can be seen in the hemangioma. The vascularization index, flow index, and vascularization–flow index were 25.4%, 42.0, and 10.7, respectively. (**d**–**f**) One month after treatment, only sparse blood vessels can be seen in the hemangioma.

**Figure 5 diagnostics-15-02903-f005:**
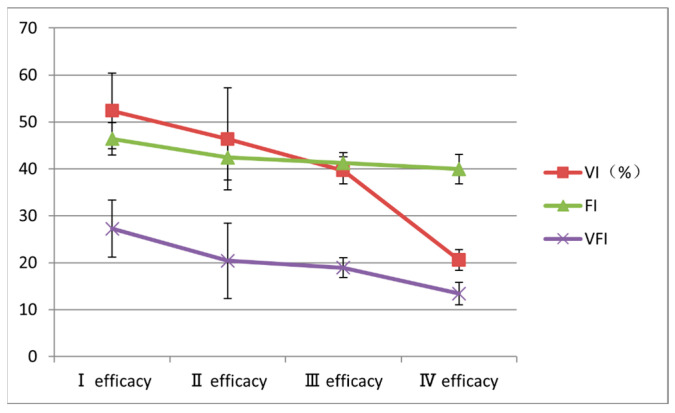
Changes in baseline three-dimensional color power angiography blood flow parameters according to treatment efficacy grade. FI, flow index; VFI, vascularization-flow index; VI, vascularization index.

**Figure 6 diagnostics-15-02903-f006:**
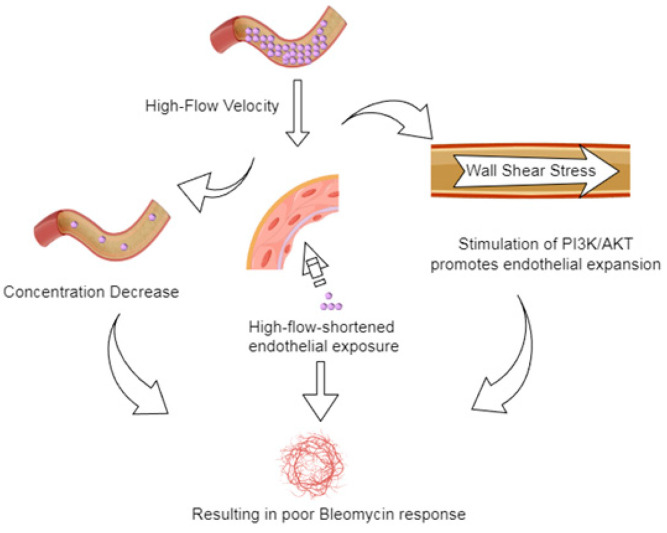
Schematic representation of the potential mechanisms by which high blood flow to lesions influence the bleomycin treatment response.

**Table 1 diagnostics-15-02903-t001:** Achauer method for evaluating infantile hemangioma treatment efficacy.

Evaluation Index	Therapeutic Grade			
	I (Poor)	II (Fair)	III (Good)	IV (Excellent)
Tumor volume reduction	0–25%	26–50%	51–75%	76–100%
Color change	Lightened	Markedly lightened	Markedly lightened	Resolved

**Table 2 diagnostics-15-02903-t002:** Baseline lesion characteristics according to treatment efficacy.

Characteristic	Treatment Efficacy Grade				χ2/Z-Value	*p*-Value
	I (*n* = 13)	II (*n* = 16)	III (*n* = 8)	IV (*n* = 3)		
**Lesion type**						
**Superficial**	0 (0%)	0 (0%)	0 (0%)	0 (0%)	0.76	1 *
**Mixed**	10 (76.92%)	12 (75%)	6 (75%)	3 (100%)		
**Deep**	3 (23.08%)	4 (25%)	2 (25%)	0 (0%)		
**Age (months)**						
**0–3**	6 (46.15%)	6 (37.5%)	3 (37.5%)	1 (33.3%)	6.45	0.736 *
**4 –6**	2 (15.38%)	7 (43.75%)	3 (37.5%)	1 (33.3%)		
**7–9**	1 (7.69%)	1 (6.25%)	1 (12.5%)	1 (33.3%)		
**10–12**	4 (30.77%)	2 (12.5%)	1 (12.5%)	0 (0%)		
**Lesion location**						
**Face**	3 (23.08%)	3 (18.75%)	3 (37.5%)	0 (0%)	4.94	0.905 *
**Head and neck**	2 (15.38%)	3 (18.75%)	0 (0%)	1 (33.3%)		
**Trunk**	7 (53.85%)	9 (56.25%)	4 (50%)	2 (66.7%)		
**Limbs**	1 (7.69%)	1 (6.25%)	1 (12.5%)	0 (0%)		

* The *p*-value derived from Fisher’s exact test.

**Table 3 diagnostics-15-02903-t003:** Comparison of three-dimensional color power angiography parameters of lesions according to treatment efficacy.

Parameter	Treatment Efficacy Grade				F-Value	*p*-Value
	I (*n* = 13)	II (*n* = 16)	III (*n* = 8)	IV (*n* = 3)		
**Volume (mL)**	7.86 ± 7.0	7.25 ± 12.12	5.45 ± 2.33	29.45 ± 41.82	2.692	0.061 *
**Vascularization index (%)**	52.35 ± 8.06	46.36 ± 10.86	39.66 ± 2.93	20.57 ± 2.22	12.472	<0.001 *
**Flow index**	46.38 ± 3.43	42.41 ± 4.75	41.21 ± 2.17	39.93 ± 3.16	4.618	0.008 *
**Vascularization–flow index**	27.23 ± 6.09	20.4 ± 8.02	18.93 ± 2.15	13.4 ± 2.42	5.746	0.003 *

* The *p*-value was derived from One-way ANOVA.

**Table 4 diagnostics-15-02903-t004:** Univariate logistic regression analysis of the association between blood flow parameters and treatment efficacy.

Parameter	B	SE	Wald	*p*-Value *	OR	95% CI
**Vascularization index**	−0.121	0.053	5.143	0.023	0.886	0.798–0.984
**Flow index**	−0.332	0.118	7.873	0.005	0.717	0.569–0.905
**Vascularization-flow index**	−0.201	0.076	7.11	0.008	0.818	0.705–0.948

CI, confidence interval; OR, odds ratio; SE, standard error. * The *p*-value derived from One-way ANOVA.

## Data Availability

The data presented in this study are available on request from the corresponding author due to privacy and ethical restrictions.

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
