# Peer review of "Quantitative Three-Dimensional Color Power Angiography Parameters Predict Response to Locally Injected Bleomycin in Infantile Hemangioma"

_diagnostics, 2025, doi:10.3390/diagnostics15222903_

Round 1

Reviewer 1 Report

Comments and Suggestions for Authors

Dear authors,

Major Comments

  • The paper provides insufficient patient demographic information such as age distribution, gender ratio, and comorbidities. It is necessary to clearly define the characteristics of the study population to demonstrate the generalizability of the findings.
  • There is no mention of the number of analysts involved, blinding procedures, or inter-rater agreement rates. This raises concerns about the subjectivity and reliability of the evaluations.
  • The criteria for selecting candidate variables and methods to address multicollinearity are unclear. In particular, the handling of highly correlated blood flow parameters requires further consideration.
  • The objectivity and reliability of the evaluation method used for assessing treatment effectiveness, as well as the appropriateness of using a 1-month early evaluation period, are not sufficiently justified.
  • Although the relationship between blood flow parameters and treatment effects is presented, the mechanisms such as drug dilution remain hypothetical. Therefore, more cautious language is recommended.
  • The study is based solely on a small, retrospective, single-center cohort without external validation or prospective studies, so clinical application should be approached with caution.

Minor Comments

  • The rationale behind dosing intervals and maximum doses, as well as details on safety monitoring, should be provided.
  • It would be helpful to add more details on the frequency, severity, and management of adverse effects.
  • Specifying concrete decision criteria such as the Youden index would improve clarity.
  • Some sections are repetitive or contain a high density of technical terms, making them difficult to read; simplification and conciseness are recommended.
  • The calculation formulas and significance of VI, FI, and VFI are somewhat complex; adding illustrations or examples would enhance understanding.
  • The possibility of variability in color change evaluations among raters should be acknowledged, along with any measures taken to address this and the limitations involved.
  • Discussing long-term effects and recurrence rates would strengthen the paper’s persuasiven

Author Response

Major Comments

Comment 1. The paper provides insufficient patient demographic information such as age distribution, gender ratio, and comorbidities. It is necessary to clearly define the characteristics of the study population to demonstrate the generalizability of the findings.

Response 1:Dear Reviewer 1: Thank you for your meticulous review of this manuscript and your valuable comments. Among the 40 children included in the study, 29 were female and 11 were male. The age at initial diagnosis, mainly recorded in the early proliferative stage, was 16 cases <3 months after birth, 13 cases between 4 and 6 months, 4 cases between 7 and 9 months, and 7 cases between 10 and 12 months. None of the children had any related complications. We have also added more clinical features of all patients in the results. We have included this information in lines 176 to 186 and changed the font color to red for your convenience.

Comment 2. There is no mention of the number of analysts involved, blinding procedures, or inter-rater agreement rates. This raises concerns about the subjectivity and reliability of the evaluations.

Response 2:Thank you very much for your valuable review comments. The analysts of this study were divided into two groups. One group was the three-dimensional ultrasound image analysis team, consisting of two ultrasound physicians with over three years of working experience, who were responsible for completing the image post-processing and generating the three-dimensional parameters of the lesion by software calculation, including the parameter values of volume V (ml), VI, FI, and VFI. The other group was the clinical medical team, consisting of two clinical doctors with over three years of experience in diagnosing and treating infantile hemangiomas, mainly responsible for evaluating the changes in the surface color of the tumor and making comparisons with the digital photo data collected before treatment.

All the analysts independently conducted all the image analysis or efficacy evaluations. Before the analysis, they all received unified training on evaluation standards. Throughout the process, strict blinding was implemented. All ultrasound images and digital photos of the tumors were anonymized and randomized. Neither the two groups of analysts nor the patients themselves knew the treatment information of the patients or the conclusions of other evaluators. We have added this explanation in lines 141 to 159 and changed the font color to red for your convenience.

Comment 3. The criteria for selecting candidate variables and methods to address multicollinearity are unclear. In particular, the handling of highly correlated blood flow parameters requires further consideration.

Response 3:We thank the reviewer for identifying this oversight in our statistical analysis. The candidate variables were selected based on literature and clinical relevance. We have now corrected this issue by using the variance inflation factor to test for multicollinearity. The VIF of VI was 4.96; the VIF of FI was 1.77, and the VIF of VFI was 5.10 (all VIF values were <10), and we used the stepwise forward regression method to screen the variables (VFI was finally retained). The details have been added to the section starting at lines 233–235 and are highlighted for your convenience.

Comment 4. The objectivity and reliability of the evaluation method used for assessing treatment effectiveness, as well as the appropriateness of using a 1-month early evaluation period, are not sufficiently justified.

Response 4:Dear Reviewer 1: Thank you for your meticulous review of this manuscript and for raising important questions regarding the scoring criteria. This study adopted the widely used Achauer efficacy assessment method in clinical practice as the efficacy evaluation standard. The Achauer standard is one of the long-standing and widely cited efficacy assessment methods in the field of hemangioma treatment. Its application ensures the comparability of the study results with previous literature. The classic Achauer standard mainly classifies based on the percentage reduction in tumor volume and appearance changes. The lesion volume is usually measured manually using a tape measure or through imaging examinations (two-dimensional ultrasound, CT, or MRI). To enhance the objectivity and accuracy of the assessment, we used the lesion volume measured by three-dimensional ultrasound as an indicator for measuring the volume change of the tumor. This measurement is more precise than the conventional results and has good repeatability, reducing the deviation from subjective judgment. Changes in the color of the tumor surface are evaluated by comparing with the digital photos collected before treatment. The evaluators are vascular tumor specialists with more than 3 years of experience in the diagnosis and treatment of infantile hemangiomas, and they have all received unified assessment standard training. The above content is presented in lines 154–159.

This study chose one month after the first treatment as the main time point for efficacy assessment mainly based on the following three reasons. Firstly, there is evidence supporting the predictive value of this early time point: for instance, Yun et al.'s study clearly set the first efficacy assessment for infantile hemangiomas one month after the injection treatment and pointed out that patients with the best efficacy at one month after hemangioma treatment maintained a better response before the treatment was terminated, whereas patients with the worst efficacy at one month did not show any further regression. This indicates that the early response can effectively predict the long-term outcome, and early assessment can effectively identify treatment responsiveness, providing a basis for timely adjustment of the plan. Secondly, this move is in line with the urgent needs of clinical practice. Given the rapid progression of proliferative hemangiomas, delayed assessment may lead to the continued progression of lesions under ineffective treatment. Conducting the assessment one month after the treatment aims to identify non-responders early and intervene promptly, which aligns with the "early, active assessment and intervention" principle emphasized by the Clinical Practice Guideline for the Management of Infantile Hemangiomas for rapidly proliferating lesions, thereby minimizing the risk of complications such as ulcers. Finally, this time point has good clinical feasibility and consensus. Setting the assessment at one month after the treatment (or 4-6 weeks) is a common practice in the field of vascular diseases, and multiple clinical studies have adopted the first efficacy assessment 4–6 weeks after the treatment, ensuring that the research design is in line with the real-world scenario. The above content is presented in lines 290–310 and is highlighted for your reference.

Comment 5. Although the relationship between blood flow parameters and treatment effects is presented, the mechanisms such as drug dilution remain hypothetical. Therefore, more cautious language is recommended.

Response 5:Thank you for your valuable suggestions. Your suggestions have made our research more rigorous. We have added "probably" at lines 337 and 338 to make the language more rigorous and standard. We will also conduct further experimental research on this mechanism.

Comment 6. The study is based solely on a small, retrospective, single-center cohort without external validation or prospective studies, so clinical application should be approached with caution.

Response 6:Thank you very much for your valuable suggestions. They have made our research more rigorous. We acknowledge the limitations of the retrospective single-center design. This study is exploratory in nature, and we will conduct multi-center prospective validation in the future. We have added this part in lines 377–384 and changed the font color to red for your easy reference.

Minor Comments

Comment 1. The rationale behind dosing intervals and maximum doses, as well as details on safety monitoring, should be provided.

Response 1:Thank you for the questions you raised regarding our research. Your suggestion has made our research more comprehensive. The selection of the dosage and treatment intervals during the treatment was based on the long-term medication experience of our institution and previous research reports. During the treatment process, we have always paid great attention to the safety issues of bleomycin application. By improving the injection technique, strictly controlling the dosage, and enhancing the efficiency of medication administration, we have attempted to minimize the single injection volume and the cumulative dosage as much as possible to reduce the incidence of adverse reactions.

Before injecting the medication, an intravenous access was opened to prepare for emergency intravenous administration in case of an emergency. During the injection treatment and after the treatment, active monitoring was carried out to promptly detect adverse reactions. The injection treatment was conducted in the examination room with rescue conditions. After the treatment, the patients were observed under monitoring for 1 hour before being allowed to leave the hospital. We have supplemented the patient treatment strategy and safety monitoring information flow chart (Figure 1) in Section 2.2. Figure 1 has been placed on line 114-115.

Comment 2. It would be helpful to add more details on the frequency, severity, and management of adverse effects.

Response 2:Thank you for your very meaningful opinion. This study included a total of 40 children. Among them, 8 cases (20.0%) had mild local swelling, which all resolved spontaneously within 48 hours and 5 cases (12.5%) developed brief low fever after treatment, which were relieved after the family members provided physical cooling. None of the children experienced serious adverse events such as injection site necrosis, ulceration, or pulmonary fibrosis.

The most dangerous adverse reaction is acute allergic reaction, which usually occurs during or shortly after the injection treatment. Although it is relatively rare, it can be life-threatening in severe cases. Therefore, the injection treatment is always conducted in a clinic with rescue conditions. After the treatment, patients need to be observed under supervision for 1 hour before being allowed to leave the hospital. None of the 40 children experienced acute allergic reactions. We have placed this part in lines 184–186 and changed the font color to red for your convenience.

Comment 3. Specifying concrete decision criteria such as the Youden index would improve clarity.

Response 3:Thank you very much for raising the statistical questions. This is a mistake on our part. The Youden index is calculated based on the ROC curve (maximizing sensitivity and specificity), with the critical value being 23.3. We have presented this information in lines 245–248 and changed the font color to red for your easy reference.

Comment 4. Some sections are repetitive or contain a high density of technical terms, making them difficult to read; simplification and conciseness are recommended.

Response 4:We sincerely thank the reviewer for pointing out the issues regarding readability and the density of technical terms. We agree that simplifying the content is crucial for conveying our message effectively. To directly address this concern, we have implemented the following two key improvements. 1. We have created a clear and concise flowchart that visual outlines the patient management strategy (figure 1, line 114-115). The flowchart provides an immediate, at-a-glance understanding of the complex process, making it much more accessible to readers without sacrificing technical accuracy. 2. We did the language polishing again to further improve the readability and conciseness of the manuscript. The entire text has undergone professional language editing by a specialized team. We believe that these revisions have substantially improved the overall clarity and readability of the manuscript, and we are grateful for the suggestion.

Comment 5. The calculation formulas and significance of VI, FI, and VFI are somewhat complex; adding illustrations or examples would enhance understanding.

Response 5:Thank you for your feedback. We have added illustrations to show the calculation process of VI/FI/VFI to enhance readability (Figure 2). We have placed the images and captions on lines 130 to 138.

Comment 6. The possibility of variability in color change evaluations among raters should be acknowledged, along with any measures taken to address this and the limitations involved.

Response 6:Thank you for your very insightful suggestions. Your opinions are absolutely correct. We must admit that this is an inevitable limitation of ours. We have adopted many methods to try to enhance the reliability and consistency of the assessment results. Firstly, the analysts for this item were all clinical doctors with more than 3 years of experience in the specialized diagnosis and treatment of infantile hemangiomas. The Achauer efficacy assessment method is the most basic assessment tool in outpatient work. The analysts have extensive clinical experience and have all received unified assessment standard training before the analysis. Secondly, the same camera equipment is used for the collection of lesion images, and the lesions are photographed in the front position under the same lighting conditions as much as possible. Even considering the above measures, the assessment results of color changes will still have a certain degree of subjectivity. We have placed this part in lines 377–380 and changed the font color to red for your convenience.

Comment 7.Discussing long-term effects and recurrence rates would strengthen the paper’s persuasiven

Response 7:Thank you very much for your valuable comments. This aspect will be explored further in future research. Currently, the research period is relatively short, and the long-term efficacy is still under follow-up observation. Due to the limited sample size, no patients with recurrence after treatment cessation have been observed yet. Moreover, some parents, due to social and economic factors, failed to visit regularly for follow-up, which also led to the loss of follow-up for some patients in the long term, affecting the completeness of the data, and the long-term data have not been included. In the future, we will maintain the continuation of this research, continuously improving the contents related to treatment responsiveness, incidence of adverse events, long-term efficacy, and recurrence rate.

Reviewer 2 Report

Comments and Suggestions for Authors

Reviewer report

Influence of the Underlying Cardiomyopathy on the Histopathological and Clinical Outcomes in a Heart Transplant Single-Center Retrospective Study

This article focuses on how the vascularization-flow index can be used as an objective predictor of the response of infantile hemangioma to locally injected bleomycin. The article is well written and describes the methods and introductions in a good manner. Articles still have scope for improvement in the results section, especially to better understand.

  1. Do the authors mention the global burden of the disease mentioned in the introduction? If not, please mention.
  2. Please describe the patient strategies involved in the form of a flow chart.
  3. What were the inclusion and exclusion criteria? Please explain.
  4. Please represent the B-mode images along with the color Doppler images in the figures.
  5. Which Scoring standards were used in the study? Please explain.
  6. Did the authors mention Clinical features before the first treatment? If no, please mention.
  7. In the references section, reference 1 is missing. Please add the reference

Author Response

This article focuses on how the vascularization-flow index can be used as an objective predictor of the response of infantile hemangioma to locally injected bleomycin. The article is well written and describes the methods and introductions in a good manner. Articles still have scope for improvement in the results section, especially to better understand.

Comment 1. Do the authors mention the global burden of the disease mentioned in the introduction? If not, please mention.

Response 1:Thank you very much for your valuable review comments. Based on your suggestions, we have rewritten the first paragraph of the introduction. Lines 45–55 of the manuscript can also be directly accessed below:

The incidence of IH is approximately 5% among live births, making it the most common benign soft tissue tumor in pediatric dermatology. Its pathogenesis is associated with the abnormal proliferation of vascular endothelial cells and neovascularization. Although most IH cases are self-limiting, some patients may develop complications such as ulcers, bleeding, functional disorders or disfigurement. In up to 20% of cases, early intervention may be strongly considered. This not only brings significant psychological stress and economic burden to the families of the affected children, but also consumes a large amount of medical resources. Moreover, there may be racial disparities in the incidence of IH; for instance, it is more commonly observed in Caucasian infants, highlighting an inequitable distribution of the disease burden.Thus, the effective management of IH underscores a matter of significant public health implications.

Comment 2. Please describe the patient strategies involved in the form of a flow chart.

Response 2:Thank you very much for your valuable suggestions. We have followed your advice and carefully drawn up a detailed flowchart. This flowchart has been added as Figure 1 to Section 2.2 of the manuscript (line 114-117).

Comment 3. What were the inclusion and exclusion criteria? Please explain.

Response 3:Dear Reviewer: Thank you for your review of this manuscript and for your valuable suggestions. Your questions regarding the inclusion and exclusion criteria are very important. We have explained them in lines 93–99 of the paper. Now, in accordance with your request, we will further elaborate as follows:

(1)Specific Inclusion Criteria

The design of the inclusion criteria is to select typical, untreated cases of infantile hemangioma that are suitable for this study. Specifically, it includes: â‘  Anatomical characteristics: The maximum diameter of a single lesion is less than 5 centimeters. Due to the limitation of the three-dimensional probe scanning area, only children with a maximum diameter of the lesion less than 5 cm are included. â‘¡Completeness of imaging data: Three-dimensional ultrasound examinations were completed before the first treatment and 1 month after the first treatment, and the images were clear and quantifiable. This is the core data source for this study, ensuring the accurate calculation of baseline parameters (VI, FI, VFI) before treatment and volume changes after treatment to evaluate the efficacy. â‘¢Treatment history: The child had not received any treatment for this hemangioma (such as medication, laser, surgery) in the past. This is to exclude the potential influence of previous treatment on the hemodynamic characteristics of the lesion and the responsiveness to this treatment, ensuring that the observed efficacy is purely due to the injection of bleomycin.

(2) Specific Exclusion Criteria (Exclusion Criteria)

The exclusion criteria are designed to minimize confounding factors, improve the reliability and accuracy of the research results, and include:

â‘ Poor treatment compliance: Cases where the treatment could not be completed due to severe adverse reactions or subjective reasons of the parents during the treatment process. Excluding such cases can avoid misjudgment of the efficacy due to treatment interruption. â‘¡Lack of follow-up data: Children who failed to return to the hospital for three-dimensional ultrasound re-examination 1 month after the first treatment as required by this study protocol. We have ensured that all included cases have complete paired data (before treatment vs. after treatment) for statistical analysis.

Comment 4. Please represent the B-mode images along with the color Doppler images in the figures.

Response 4:Thank you very much for your valuable suggestions. We have made the necessary changes according to your advice and have added the patient's ultrasound images and color Doppler images to lines 192–205 of the manuscript.

Comment 5. Which Scoring standards were used in the study? Please explain.

Response 5:Dear Reviewer 2: Thank you for your meticulous review of this manuscript and for raising important questions regarding the scoring criteria. We have explained the scoring criteria used in paragraphs 140-144 of the paper. Now, in response to your request, we will provide a more detailed explanation.

(1) Selection and Basis of Scoring Criteria

This study adopts the widely used Achauer efficacy assessment method as the core efficacy evaluation standard. The selection of this standard is based on the following two key considerations:

Recognizability and Comparability: The Achauer standard is one of the long-standing and widely cited efficacy assessment methods in the field of hemangioma treatment. Its application ensures the comparability of the results of this study with previous literature.

Applicability and Quantifiability: The classic Achauer standard mainly assesses the tumor volume reduction percentage and appearance changes for grading. The tumor volume is usually measured manually using a tape measure or through imaging examinations (two-dimensional ultrasound, CT, or MRI). To enhance the objectivity and accuracy of the assessment, we use the lesion volume measured by three-dimensional ultrasound as an indicator of tumor volume change. This measurement is more precise than the conventional results and has better repeatability, reducing the deviation from subjective judgment. Changes in the surface color of the tumor are evaluated by comparing images with the digital photos collected before treatment. The evaluators are vascular tumor specialists with more than three years of experience in the diagnosis and treatment of infantile hemangiomas, and they have received unified assessment standard training.

(2) Specific Content and Application of the Achauer Scoring Criteria

The efficacy is classified into 4 grades (I-IV), and the specific grading content is as follows:

Evaluation index

Therapeutic grade

Ⅰ(Poor)

Ⅱ(Fair)

Ⅲ(Good)

Ⅳ(Excellent)

The tumor volume reduction

0-25%

26-50%

51-75%

76-100%

Color change

lightening

marked lightening

marked lightening

resolution

Comment 6. Did the authors mention Clinical features before the first treatment? If no, please mention.

Response 6:Thank you very much for your valuable suggestions. You have pointed out an important issue regarding the description of clinical characteristics before the first treatment. According to your suggestion, we have provided additional explanations and placed this part in lines 177 to 184, with the text in red font, to facilitate review. You can also directly access it below:

A total of 40 patients received local injection treatment. The cohort consisted of 29 women (72.5%) and 11 men (27.5%). The age distribution at the initial diagnosis was as follows: 16 patients (40.0%) were 0–3 months old; 13 cases (32.5%), 4–6 months old; 4 cases (10.0%), 7–9 months old; and, 7 cases (17.5%), 10–12 months old. The trunk was the most common location of the lesion, with 22 cases (55.0%), followed by the face with 9 cases (22.5%), the head and neck with 6 cases (15.0%), and the limbs with 3 cases (7.5%). In terms of lesion type, the majority were mixed types (31 cases, 77.5%) and deep types (9 cases, 22.5%). No superficial type lesions were observed in this cohort.

Comment 7. In the references section, reference 1 is missing. Please add the reference

Response 7:Thank you very much for your meticulous review of our manuscript and for pointing out the issue of the absence of the first citation in the reference list. This was indeed a flaw in our literature management process, and we sincerely apologize for it. In accordance with your request, we have conducted a verification and made the necessary additions (lines 399–400).

Reviewer 3 Report

Comments and Suggestions for Authors

Infantile hemangioma is always a great burden for parents, but also for doctors. Although treatment with beta-blockers has yielded good results, according to this review, treatment with bleomycin may become a treatment method. Although IH is basically a benign tumor, it is still a consequence of abnormal proliferation of vascular endothelial cells, and bleomycin, according to the exact results of this study, leads to tumor regression with local application once a month and without visible adverse events. The use of local injection with measurement of the vascularization index, flow index and vascularization flow index proves the beneficial effect of this method in the regression of IH. The excellent presentation of the results by three-dimensional color angiography and the application of ROC analysis in the presentation of the results contribute significantly to the quality of this study. Congratulations.

Author Response

Dear Reviewer:

After reading your highly positive review, all of us as authors feel extremely honored and express our most sincere gratitude. You accurately summarized the core findings of this research and fully affirmed our attempts in objective assessment indicators such as the vascularization index, three-dimensional vascular imaging, and ROC analysis. This provided us confidence and direction for our subsequent research.

Your encouragement is the greatest motivation for us to move forward. We will definitely carefully absorb your affirmation and continue to strive on the basis of our current work, endeavoring to achieve more solid results in this field. Once again, thank you for your time spent on reviewing and your generous encouragement!

Round 2

Reviewer 1 Report

Comments and Suggestions for Authors

Dear Authors,

I am satisfied with the revisions that have been made by the authors.

Author Response

Comment: I am satisfied with the revisions that have been made by the authors.
Response: We are pleased to hear that you are satisfied with our revisions. We thank you once again for your insightful comments and guidance, which have significantly strengthened our manuscript.